# Rapid Diagnostic of *Streptococcus suis* in Necropsy Samples of Pigs by *thrA*-Based Loop-Mediated Isothermal Amplification Assay

**DOI:** 10.3390/microorganisms11102447

**Published:** 2023-09-29

**Authors:** Julian Hess, Antonia Kreitlow, Karl Rohn, Isabel Hennig-Pauka, Amir Abdulmawjood

**Affiliations:** 1Field Station for Epidemiology (Bakum), University of Veterinary Medicine Hannover, Foundation, 49456 Bakum, Germany; julian.hess@tiho-hannover.de; 2Institute for Food Quality and Food Safety, University of Veterinary Medicine Hannover, Foundation, 30173 Hannover, Germany; antonia.kreitlow@tiho-hannover.de; 3Institute for Biometry, Epidemiology and Information Processing, University of Veterinary Medicine Hannover, Foundation, 30559 Hannover, Germany; karl.rohn@tiho-hannover.de

**Keywords:** *S. suis*, *thrA* gene, LAMP, brain, joint

## Abstract

*Streptococcus* (*S.*) *suis* presents a serious threat to the pig industry as well as food safety and public health. Although several LAMP assays have been developed for the identification of *S. suis*, no universal assay is so far available for the field-suitable examination of clinical pig specimens. Based on the *thrA* housekeeping gene, a new loop-mediated isothermal amplification (LAMP) assay was developed and validated for the detection of *S. suis* in the brain and joints of pigs. For this LAMP assay, two different methods for the extraction of DNA from brain and joint swabs were compared. Using the LPTV boiling method, the detection limit of LAMP was 1.08 CFU/reaction, while the detection limit was 53.8 CFU/reaction using a commercial DNA extraction kit. The detection limits of *thrA*-LAMP in combination with the LPTV boiling method were 10^4^–10^5^ CFU/swab in the presence of brain tissue and 10^3^–10^4^ CFU/swab in the presence of joint tissue. The diagnostic quality criteria of LAMP were determined by the examination of 49 brain swabs and 34 joint swabs obtained during routine diagnostic necropsies. Applying the LPTV boiling method to brain swabs, the sensitivity, specificity, and positive and negative predictive values of *thrA*-LAMP were 88.0, 95.8, 95.7, and 88.5% using cultural investigation as a reference method, and 76.7, 100, 100, and 73.1% using real-time PCR as a reference method. Based on these results, the *thrA*-LAMP assay combined with the LPTV boiling method is suitable for rapid detection of *S. suis* from brain swabs.

## 1. Introduction

*Streptococcus* (*S.*) *suis* is a gram-positive bacterium that inhabits the upper respiratory tract, specifically the tonsils and nasal cavities, of pigs [1]. Based on the capsular polysaccharides (CPS), *S. suis* strains have been serologically classified into 35 serotypes (1–34 and 1/2) until the end of the last century [2]. However, due to new phylogenetic analyses, serotypes 20, 22, 26, and 32–34 have been recently allocated to other species [3,4]. Furthermore, serotypes 20, 22, and 26 were proposed as a novel species called *Streptococcus parasuis* sp. nov. [5]. A recent study suggested that *S. parasuis* and *S. suis* still share genetic characteristics, and even horizontal gene transfer is considered possible [6]. Serotype 33 was reclassified as the new species *Streptococcus ruminantium* sp. nov. [7]. *S. suis* serotypes 32 and 34 reference strains were assigned to the species *S. orisratti* [8]. Investigation of the epidemiological situation of *S. suis*-serotypes in German pig herds in 2019 by Prüfer et al. revealed serotypes 2 and 1/2, 9, 7, 4, 1, and 14 to be the most common within the more recent collection of invasive isolates [9]. Data on global distribution showed that serotypes 2, 9, 3, 1/2, and 7 were predominant in clinically infected pigs [10]. Despite enormous progress in research focusing on virulence factor genes, the pathogenesis of a systemic *S. suis* infection is still not fully understood [11]. *S. suis* might cause disease when different biotic or abiotic predisposing factors, such as virus infections or stress, impact the host’s immunity [1,12]. In case of an outbreak, typical clinical signs are meningitis, arthritis, endocarditis, pneumonia, or peracute death [1,12]. Outbreaks most frequently occur in nursery piglets, but also suckling piglets can be affected [1]. Without antibiotic treatment, mortality can rise up to 20–30% [1,13]. Mortality during the nursery phase can lead to relevant economic losses [14]. *S. suis* is also a zoonotic pathogen, with most cases reported in China and Southeast Asia, mainly by serotype 2 [10,15]. Sporadic cases of *S. suis* infections in humans have also been reported in other regions of the world [10,15]. Humans in close contact with pigs or pork can get infected through minor skin lesions [12]. Consumption of raw pork products is also considered a possible route of infection [16]. Similar to pigs, human infections often result in meningitis or other manifestations like streptococcal toxic shock-like syndrome, septicaemia, arthritis, pneumonia, and peritonitis [12,17]. Therefore, *S. suis* represents a serious threat, not only to livestock and the economy but also in terms of food safety and public health [14,15,16]. Rapid and reliable diagnosis of this pathogen and its diversity of serotypes in various different samples is crucial. Detection of *S. suis* by standard culture methods is laborious, and biochemical characteristics may be inconclusive [18,19]. PCR-based assays show high sensitivity and specificity but require expensive equipment and are not suitable for on-site operations [20,21]. In contrast, loop-mediated isothermal amplification (LAMP) proved to be a suitable technique to overcome these shortcomings [21,22,23].

Since the LAMP reaction is performed under isothermal conditions, a simple water bath or heating block can be used [24]. On this basis, portable devices have been developed, allowing the method to be performed directly on-site [25]. In previous studies, it was shown that LAMP is less impacted by the effects of the biological matrix than PCR, suggesting that the step of DNA purification could be omitted for a LAMP assay [25,26,27,28]. This would facilitate the use of LAMP for diagnostics outside of laboratories [26]. Previously developed serotype-specific LAMP assays enabled the detection of highly pathogenic *S. suis* serotype 2 and 14 strains in samples of infected human patients [21,29]. The *recN*-based LAMP by Arai et al., could correctly identify all serotypes, except those recently excluded from the species *S. suis* [23]. Using these LAMP assays, the prevalence of *S. suis* in raw pig samples in Thailand was investigated [30]. Results revealed deficiencies in food safety due to cross-contamination with *S. suis* during meat handling, especially in local retail stores and markets [30]. Although several LAMP assays have been developed focusing on *S. suis* with regard to human public health and food safety, no systems have been established to detect infections in pigs. An appropriate diagnostic LAMP assay would facilitate early detection of *S. suis* outbreaks in pig herds. Consequently, antibiotic treatment could be initiated quickly, reducing mortality and economic loss as well as preventing further infection of humans in close contact [12,14]. The present study aimed at developing and validating a field-suitable LAMP assay for detecting the broad spectrum of *S. suis* variants with high specificity and sensitivity in the brain and joints of infected pigs.

## 2. Materials and Methods

### 2.1. Bacterial Strains

A total of 220 bacterial strains were examined in this study. Reference strains originated from varying culture collections such as the American Type Culture Collection (Manassas, VA, USA), the Culture Collection University of Gothenburg (Göteborg, Sweden), the DSMZ-German Collection of Microorganisms and Cell Cultures GmbH (Leibniz Institute, Braunschweig, Germany), and the National Collection of Type Cultures (Salisbury, UK). Field strains were provided by the Field Station for Epidemiology in Bakum (University of Veterinary Medicine Hannover, Foundation, Hannover, Germany) and the Institute for Food Quality and Food Safety (University of Veterinary Medicine Hannover, Foundation, Hannover, Germany). The *S. suis* field strains used in this study were obtained during internal routine diagnostics from various pig tissues. The *S. suis* isolates were recovered by standard culture methods and subsequently confirmed and serotyped by PCR (see Section 2.10). Detailed biochemical identification using the API^®^ 20 STREP system (bioMérieux, Marcy-I’Etoile, France) was performed and interpreted according to the manufacturer’s manual. Field strains of non-target species were isolated in the respective institutes by standard culture methods during routine diagnostics, and confirmed via biochemical characterization or MALDI-TOF MS analysis. All bacterial strains were stored in cryobanks (Mast Diagnostica GmbH, Reinfeld, Germany) at −80 °C until further use.

### 2.2. Extraction of Genomic DNA from Bacterial Cultures

*Glaesserella parasuis*, *Histophilus somni*, and *Actinobacillus* spp. were plated out on Chocolate Agar (Blood Agar No. 2 Base, Becton Dickinson GmbH, Heidelberg, Germany) and incubated for 24 h at 36 °C under microaerobic (≥2.5% CO_2_) and aerobic conditions, respectively. *Actinomyces hyovaginalis* (36 °C), *Brachyspira* spp. (36 °C), and *Clostridium pefringens* (45 °C) isolates were cultured on Columbia Agar containing 5% sheep blood (Becton Dickinson GmbH, Heidelberg, Germany) at anaerobic conditions (≥13% CO_2_) for 24–48 h. For cultivation of *Campylobacter jejuni* and *Campylobacter coli*, strains were incubated on Columbia agar with sheep blood under microaerobic conditions at 42 °C for 24–48 h. All other strains, including *S. suis*, were cultured on Columbia sheep blood agar (Becton Dickinson GmbH, Heidelberg, Germany) under aerobic conditions at 36 °C for 24–48 h. Genomic DNA of five to ten colonies of each strain was extracted using the DNeasy^®^ Blood & Tissue Kit (Qiagen GmbH, Hilden, Germany) according to the manufacturer‘s manual. The DNA concentration of each eluate was measured in duplicate using the Thermo Scientific^TM^ Multiskan^TM^ GO spectrophotometer together with the Thermo Scientific^TM^ μDrop^TM^ Plate and calculated using the SkanIt^TM^ Software Version 4.1 (Thermo Fisher Scientific Corporation, Waltham, MA, USA). For standardization of running conditions in analytical specificity testing, each eluate was adjusted to a DNA concentration of 0.1 ng/μL.

### 2.3. Design of LAMP Primers

LAMP primer design was based on the nucleotide sequence of the *thrA* gene (Acc No.: DQ205250.1). Corresponding sequence data was obtained from the database of the National Center for Biotechnology Information (NCBI) (Bethesda, MD, USA). A set of six primers, including forward and backward outer primers (F3 and B3), forward and backward inner primers (FIP and BIP), as well as forward and backward loop primers (LF and LB), for accelerating amplification, was created using LAMP Designer software (OptiGene Ltd., Horsham, UK) [31]. Primer sequences are shown in Table 1. All LAMP primers were ordered from Eurofins Genomics Germany GmbH (Ebersberg, Germany).

### 2.4. LAMP Reaction

LAMP reactions were carried out as suggested in the OptiGene Ltd. LAMP User Guides (Version 1.1, OptiGene Ltd., Horsham, UK). For each LAMP reaction, a 25-µL reaction mixture containing 15 μL of GspSSD isothermal mastermix (ISO-001) (OptiGene Ltd.), 2.5 μL primer mix, 2.5 μL PCR-grade water (Qiagen GmbH, Hilden, Germany), as well as 5 μL template, was prepared. LAMP reactions were performed using the real-time fluorometer Genie^®^ II (OptiGene Ltd., Horsham, UK).

### 2.5. Optimization of the LAMP Assay

Optimization of the LAMP assay was performed in accordance with OptiGene Ltd. LAMP User Guides (Version 1.1, OptiGene Ltd., Horsham, UK). At first, a standard and a concentrated primer mix version were prepared. Single primer concentrations per reaction were 0.2 µM F3/B3, 0.8 µM FIP/BIP, and 0.4 µM LF/LB using the standard primer mix, and 0.2 µM F3/B3, 2.0 µM FIP/BIP, and 1.0 µM LF/LB using the concentrated primer mix. The optimum reaction temperature of both primer mixes was determined by testing *S. suis* reference strain (DSM 9682) DNA (0.1 ng/µL) at different temperatures ranging from 62 to 69 °C (∆*T* = 1 °C). Each run consisted of a 40-min amplification phase followed by an annealing period to generate a melting curve in the range from 98 to 80 °C (ramp rate 0.05 °C/s). Considering the standard deviation of all measurements, the shortest mean detection time determined the individual optimal reaction temperature of the standard and concentrated primer mixes. Subsequently, the most suitable primer mix version was selected based on analytical sensitivity and the corresponding detection times. For this purpose, a ten-fold series dilution of *S. suis* reference strain DNA was prepared with concentrations ranging from 10 ng/µL to 10 fg/µL. Each dilution stage served as a template for the LAMP reaction using the standard and concentrated primer mix at the previously determined optimal reaction temperature. The amplification and annealing phases of each run were set as described above. The lowest DNA concentration that could be successfully detected in triplicate determined the analytical sensitivity of the primer mix. All optimization experiments were performed three times in a row.

For all subsequent runs, the instrument settings were chosen as described above. Furthermore, each run included one reaction with a 5 µL template of *S. suis* DSM 9682 DNA (0.1 ng/µL) as a positive control and one reaction using 5 µL PCR-grade water instead of the DNA template as a negative control.

### 2.6. Analytical Specificity Testing

To investigate the analytical specificity of the *thrA*-LAMP assay, DNA from a total of 220 bacterial reference strains and field isolates was tested. Inclusivity testing was performed using 104 *S. suis* isolates. In order to examine as many genetically distinct *S. suis* strains as possible, isolates of every available serotype were chosen. Within the same serotype, isolates were selected to represent every available pattern of virulence-associated factor genes. Isolates from as many different tissues and herds as possible were selected within a combination. Exclusivity was verified by testing 116 non-*S. suis* isolates. Non-target isolates were selected based on their close genetic relationship with *S. suis* sp. or similar clinical and pathomorphological characteristics. Species that share the pig as a host or occur in its environment were also of interest. To confirm the reaction specificity, a melting curve was generated for each reaction as described above.

### 2.7. Determination of the Bacterial Cell-Based Detection Limit Using Different DNA Extraction Methods

The cell-based detection limit of the assay was determined in triplicate using two different methods for extracting DNA. For this purpose, the *S. suis* reference strain (DSM 9682) was cultured on Columbia sheep blood agar (Becton Dickinson GmbH, Heidelberg, Germany) under aerobic conditions at 36 °C for 48 h. Colonies of the reference strain were suspended in 5 mL of 0.9% isotonic saline solution until a turbidity of 1.0 McFarland units (MFU) was measured by a densitometer (Densimat, bioMérieux, Marcy-I’Etoile, France). This cell suspension was ten-fold serially diluted with isotonic saline solution. Viable cell counts were determined using the plate count technique as described previously [32]. Culture conditions of incubated Columbia sheep blood plates were adjusted for *S. suis* to 36 °C for 48 h under aerobic conditions. DNA was extracted from each cell suspension using two distinct procedures.

For thermal extraction, 50 µL of each dilution was transferred to 450 µL of variplex LPTV buffer (AmplexDiagnostics GmbH, Gars Bahnhof, Germany) in a 2 mL microcentrifuge tube. Subsequently, each tube was mixed by vortexing and boiled for 10 min at 100 °C on the Eppendorf ThermoMixer^®^ C (Eppendorf AG, Hamburg, Germany). After cooling down to room temperature for 10 min, the inoculated LPTV buffer was directly used as a template for LAMP. This procedure is further referred to as the LPTV boiling method. Enzymatic extraction and DNA purification were performed using the DNeasy^®^ Blood & Tissue Kit (Qiagen GmbH, Hilden, Germany). One milliliter of each dilution was transferred to a 1.5 mL microcentrifuge tube and centrifuged at 10,000× *g* for 5 min. After discarding the supernatant, the bacterial pellet was resuspended in 180 μL enzymatic lysis buffer, mixed by vortexing, and incubated on a thermoshaker (Thermomixer Comfort, Eppendorf AG, Hamburg, Germany) at 37 °C for 1 h (1000 rpm). Subsequently, 25 μL of Proteinase K and 200 μL of Buffer AL were added to each tube and mixed by vortexing. The mixture was incubated at 56 °C for 1 h (1000 rpm). All further steps were carried out according to the manufacturer’s instructions (Qiagen GmbH, Hilden, Germany). The elution volume was set to 200 μL. Each eluate served as a template for the LAMP reaction.

### 2.8. Real-Time PCR Assay

Detection of *S. suis* by real-time PCR was performed using the BactoReal^®^ Kit *Streptococcus suis* (ingenetix GmbH, Vienna, Austria) based on the fibrinogen binding protein gene (*fbpS*). The real-time PCR was performed according to the kit manufacturer’s instructions using the Applied Biosystems 7500 Real-Time PCR System (Life Technologies GmbH, Darmstadt, Germany). Ct-values were interpreted according to the kit manufacturer’s instructions.

### 2.9. Detection Limits of LAMP in Artificially Contaminated Brain and Joint Samples

The preparation of a tenfold dilution series up to dilution level 10^−8^ of the *S. suis* DSM 9682 strain and the determination of viable cell counts were conducted as described above. Brain and joint tissue of pigs without clinical signs of a *S. suis* infection were sampled during routine necropsy using sterile, PCR inhibitor-free viscose swabs (nerbe plus GmbH & Co. KG, Winsen/Luhe, Germany). In dead pigs, the head was separated from the body, the scalp removed, and the cranial cavity opened by removing the skullcap using a disinfected saw. Subsequently, a swab sample was flat-inserted through the leptomeninx in the superficial brain tissue of both cerebral hemispheres. The joints were opened using a clean knife. Through an incision parallel to the joint space, the superficial skin and surrounding supporting tissue were separated, so that the joint cavity was exposed without touching the articulating surfaces. After careful insertion of a swab in the joint cavity, synovia was collected and the synovialis was brushed. Due to easy accessibility, the elbow, or alternatively, the tarsal joint, was sampled. As a sample matrix, two brain or joint swabs per prepared cell dilution level were taken. In addition, one swab per sampled brain or joint was provided to ensure the absence of *S. suis* via the cultural investigation as described below. Samples that did not produce any *S. suis* suspecting colonies on agar plates were selected for artificial contamination experiments.

Using the LPTV boiling method for DNA extraction, eight swab samples per sample type were rotated and squeezed out in one tube containing 450 µL of LPTV buffer. Subsequently, 50 µL of *S. suis* cell suspension was taken from each dilution level and added to the prepared LPTV buffer. Further steps were carried out as described previously. For DNA extraction with the DNeasy^®^ Blood & Tissue kit, the remaining eight swab samples per sample type were suspended in 500 μL of enzymatic lysis buffer in a microcentrifuge tube, followed by a vortexing step. Subsequently, 1 mL of each dilution level was transferred to a 1.5 mL microcentrifuge tube and centrifuged at 10,000× *g* for 5 min. The supernatant was carefully discarded. The cell pellet was resuspended with 180 μL of the enzymatic lysis buffer, including tissue components, and again mixed by vortexing. Further extraction was performed as described previously. Templates were tested using LAMP and real-time PCR as described previously. Experiments were performed in triplicate.

### 2.10. Detection of S. suis in Field Samples

Pigs with clinical or pathological evidence of a systemic *S. suis* infection were selected for the collection of clinical field samples. The brain and joints of these animals were sampled during diagnostic necropsies, as described above. For each brain and joint sampled, one swab was used for cultural investigation. For this purpose, the swab sample was fractionally smeared on Columbia agar containing 5% sheep blood (Becton Dickinson GmbH, Heidelberg, Germany), Chocolate Agar (Blood Agar No. 2 Base, Becton Dickinson GmbH, Heidelberg, Germany), Columbia CNA agar with 5% sheep blood (Becton Dickinson GmbH, Heidelberg, Germany), and Gassner agar (Oxoid Deuschland GmbH, Wesel, Germany), followed by incubation under aerobic or microaerobic (Chocolate agar) conditions at 36 °C for 24 h until first inspection and a further aerobic incubation until 48 h. Potentially pathogenic bacteria were selected based on their morphology, color, odor, media-specific growth, and haemolysis behavior and identified by using a specific biochemical substrate test series. In the case of the detection of *S. suis*, isolates were further cryobanked and serotyped using in-house PCR with modifications by Silva et al. and Kerdsin et al. (refer to Appendix A) [33,34]. This conventional multiplex PCR was based on the *gdh*-gene and was able to differentiate the five capsular serotypes (*cps*-types) 1, 2, 4, 7, and 9 [33,34]. In addition, virulence-associated factors were determined as genes for the muramidase-released protein (*mrp*), suilysin (*sly*), extracellular factor (*epf*), and arginine deiminase (*arcA*) [33,34]. The PCR was suitable to identify the most common invasive serotypes in Germany [9]. All isolates carrying *gdh* or at least one virulence-associated factor gene without any of the five *cps*-types are further referred to as “non-typable *S. suis*”.

Two additional swabs per sampled brain or joint were stored at minus 20 °C until DNA extraction. For extraction using the LPTV boiling method, one tissue swab was suspended in 500 μL LPTV buffer and further processed as described above. For enzymatic extraction using the DNeasy^®^ Blood & Tissue Kit, a tissue swab was suspended in 500 μL of isotonic saline solution, vortexed, and centrifuged at 10,000× *g* for 5 min. Subsequently, the supernatant was discarded, and the pellet was resuspended in 180 µL of enzymatic lysis buffer. Further steps were performed as described previously. Both DNA templates were examined by LAMP and real-time PCR. Cultural investigation with subsequent PCR-based serotyping as well as real-time PCR after DNeasy^®^ Blood & Tissue Kit extraction served as reference methods. In addition, cryobanked *S. suis* isolates from clinical samples were cultured as described above. Using a sterile inoculation loop, 5 to 10 colonies from pure subcultures were suspended in 500 μL of LPTV buffer. DNA was extracted using the LPTV boiling method and all templates were examined by *thrA*-LAMP.

### 2.11. Data Management and Statistical Analyses

Original data from LAMP and real-time PCR were viewed and processed using the Genie^®^ Explorer software (Version 2.0.7.11, OptiGene Ltd., Horsham, UK) and the 7500 Software for 7500 and 7500 Fast Real-time PCR Systems (Version 2.3, Life Technologies GmbH, Darmstadt, Germany), respectively. Statistical analyses were accomplished using Microsoft Office Excel 2016 (Microsoft Corporation, Redmond, WA, USA). Sensitivity and specificity, as well as positive and negative predictive values, each with 95% confidence intervals, were calculated using SAS 9.4m7 with the SAS Enterprise Guide, version 7.1 (SAS Institute Inc., Cary, NC, USA). The calculation of diagnostic quality criteria was performed as suggested by the U.S. Food and Drug Administration (FDA) [35].

## 3. Results

### 3.1. Optimized LAMP Reaction Conditions and Analytical Sensitivity

As the first step for assay optimization, the optimal reaction temperature of the standard and concentrated primer mixes was determined to be 65 °C for both primer mixes (fastest mean detection times, minimal standard deviation). At this temperature, a positive fluorescence signal was measured on average after 10.38 min when using the standard primer mix and after 7.83 min when using the concentrated primer mix. DNA-based analytical sensitivity was determined as 0.5 pg DNA per reaction mixture for both primer mixes. In contrast to the standard primer mix, even 50 fg of reference strain DNA could be successfully detected in 2 of 3 replicates when using the concentrated primer mix. In addition, the concentrated primer mix achieved overall faster mean detection times across all dilution levels. Considering these results, all further LAMP reactions were performed using the concentrated primer mix at a reaction temperature of 65 °C. Samples were considered positive for *S. suis* if both a fluorescence signal and a specific melting temperature of 84.63 ± 1 °C were detectable.

### 3.2. Analytical Specificity of thrA-Based LAMP

Among all 104 *S. suis* strains tested in this study, over 99% of isolates showed a positive fluorescence signal when performing the *thrA*-based LAMP. The diversity of serotypes and virulence-associated factor genes of these isolates is shown in Table 2. The isolates originated from clinically relevant organs (brain and cerebrospinal fluid, joint, heart, serosa, kidney, lung), as well as from naturally colonized tissues (bronchus, tonsil). The only *S. suis* isolate that showed no amplification by *thrA*-LAMP was identified as serotype 7, unusually without any of the detectable virulence-associated factor genes. Detailed biochemical identification using the API^®^ 20 STREP system (bioMérieux, Marcy-I’Etoile, France) determined this isolate as *S. suis* biotype II with “good identification” (95.2 percentage of identification and T = 0.64). Further MALDI-TOF MS analysis of this strain at SAN Group Biotech Germany GmbH (Höltinghausen, Germany) identified this isolate as *Streptococcus* sp. Positive fluorescence signals for the other strains were measured within 7.17 to 11.60 min after starting the amplification process. Melting temperatures of corresponding LAMP products ranged from 84.19 °C to 84.89 °C, indicating species-specific fluorescence signals. All of the 116 non-*S. suis* strains (Table 3) were tested negative using *thrA*-LAMP.

### 3.3. Bacterial Cell-Based Detection Limit of thrA-LAMP

Bacterial cell-based detection limits were determined by the lowest mean viable cell count in a ten-fold serial dilution of *S. suis* strain DSM 9682 that could be successfully detected in triplicate. Using the LPTV boiling method for DNA extraction, a mean detection limit of 1.08 CFU/reaction (2.2 × 10^3^ CFU/mL) was achieved. Kit extraction resulted in a higher mean detection limit of 53.8 CFU/reaction (2.2 × 10^3^ CFU/mL).

### 3.4. Evaluation of Brain and Joint Matrix Effects on the Detection Limit of thrA-LAMP

Detection limits of the *thrA*-based LAMP after DNA extraction using the LPTV boiling method and DNeasy^®^ Blood & Tissue Kit were determined by spiking experiments on artificially contaminated brain and joint swabs (Table 4). The detection limits were determined by the lowest mean viable cell count that could be successfully detected in triplicate. A detection limit of 10^4^–10^5^ CFU per swab was achieved by LAMP when DNA was extracted using the LPTV boiling method on brain swabs. When the same templates were examined by real-time PCR, almost no amplification of the target or the internal positive control was observed (Appendix A). DNA extraction using the DNeasy^®^ Blood & Tissue Kit improved the detection limit of *thrA*-LAMP and real-time PCR to 10^3^–10^4^ CFU/swab. Performing the LPTV boiling method on spiked joint swabs, the detection limit of LAMP was 10^3^–10^4^ CFU/swab. By applying kit extraction to joint swabs, LAMP became even less sensitive, with a detection limit of 10^4^–10^5^ CFU per swab. Compared to real-time PCR, LAMP had an at least 10-fold higher detection limit when examining joint samples.

### 3.5. Diagnostic Quality Criteria for Detection of S. suis in Brain and Joint Swabs

Diagnostic quality criteria of *thrA*-LAMP compared to cultural investigation with subsequent PCR-based serotyping as well as real-time PCR after DNeasy^®^ Blood & Tissue Kit extraction are summarized below (Table 5 and Table 6). The brains of 49 pigs were examined. *S. suis* was successfully isolated from 25 brains. 30 brain samples were positive in a real-time PCR examination. Using the LPTV boiling method, the *thrA*-based LAMP assay was able to correctly identify 22 of the culture-positive and 23 of the culture-negative brain samples. When real-time PCR served as a reference method, *thrA*-LAMP correctly identified 23 of the PCR-positive and all PCR-negative brain samples. When brain swabs were extracted and purified using the DNeasy^®^ Blood & Tissue Kit, all culture-positive samples were correctly identified by LAMP, but the number of correctly detected culture-negative samples decreased to 21. Twenty-eight PCR-positive brain samples were identified as such by *thrA*-LAMP using kit extraction. The detection time median of LAMP was 18.45 min when using the LPTV boiling method. Applying kit extraction reduced the detection time median to 10.68 min (e.g., Figure 1a,b).

Joint swabs were obtained from 34 different pigs. Of these, ten swabs were positive in the cultural investigation, and 23 joint swabs were tested positive by real-time PCR. In combination with the LPTV boiling method, *thrA*-LAMP was only able to correctly identify one joint swab as *S. suis* positive. As already observed for brain swabs, DNA extraction and purification using the DNeasy^®^ Blood & Tissue Kit increased LAMP sensitivity to 100% (cultural investigation) and 82.61% (real-time PCR), respectively. Performing kit extraction, the specificity of *thrA*-LAMP was lower when examining culture-negative (58.33%) than PCR-negative joint swabs (90.91%). The one joint sample detected by LAMP using the LPTV boiling method showed a positive fluorescence signal after 18.63 min. The detection time median was 11.63 min when kit extraction was performed.

All 35 *S. suis* isolates retrieved from clinical samples gave a positive result when performing *thrA*-LAMP when DNA was directly extracted from bacterial colonies using the LPTV boiling method. The detection time median for a positive fluorescence signal was 6.28 min. The melting temperatures of all LAMP products were specific and ranged between 84.34 and 85.07 °C.

## 4. Discussion

In this study, the first *S. suis*-specific LAMP assay was developed and validated for rapid detection of this pathogen in brain and joint samples of clinically infected pigs. The assay was based on the *thrA* gene encoding the enzyme aspartokinase/homoserine dehydrogenase [36]. Since King et al. referred to *thrA* as a housekeeping gene, this region can be considered to be a highly conserved and specific genomic target for the *S. suis* species [36]. In the present study, the *thrA*-LAMP was able to successfully detect 99% of the *S. suis* strains, representing the globally predominant serotypes found in pigs [10]. Furthermore, no cross-reactions with non-target species were observed. By standardizing the template DNA concentration to 0.1 ng/μL, varying detection times of *S. suis* strains were caused mainly by polymorphisms of the *thrA* gene sequence. Considering the given range of detection times, it can be assumed that the *thrA*-sequence shares high similarity within the *S. suis* species. The only *S. suis* strain that was not detected by *thrA*-LAMP was biochemically confirmed using the API^®^ 20 STREP system. Nevertheless, further MALDI-TOF MS analysis was inconclusive, as it identified this strain as *Streptococcus* sp. Multitest systems, such as API^®^, have been reported to misidentify *S. suis* strains, resulting in false-negative results [37,38]. MALDI-TOF MS analysis, on the other hand, tended to give false-positive results, especially in the classification of tonsil isolates [39]. Since the results of this study did not reflect any of the described cases, it seemed that the investigated strain was not a typical representative of the *S. suis* species [3]. Average nucleotide identity (ANI) analysis could be used for confident identification of this isolate [39]. Moreover, no reliable prediction can be made about its virulence and clinical relevance as this strain was isolated from the tonsil of a healthy carrier [40]. Consequently, the suitability of *thrA*-LAMP for the detection of clinically relevant *S. suis* strains remained unrestricted. The previous *S. suis*-LAMP assays developed by Zhang et al. and Meng et al. only targeted serotype-specific genes without covering the worldwide serotype distribution found in diseased pigs and humans [10,21,29,41]. While the *recN*-based LAMP developed by Arai et al. was able to detect all serotypes according to the currently valid taxonomy, a smaller number of 54 *S. suis* strains, as well as 19 non-target strains, were used for analytical specificity testing. Furthermore, for most serotypes, only one representative strain was tested without considering virulence-associated factor genes, so the data on inclusivity may not be entirely valid [23]. By including twice as many *S. suis* isolates and covering multiple representatives per serotype with distinct virulence-associated factor gene patterns, the *thrA*-LAMP was successfully tested on a population with higher genetic variability.

For the application of LAMP under field conditions, a fast and easy-to-use DNA extraction method was required [42]. The developed LPTV boiling method can be performed with a heating block in less than 20 min, thus requiring only minimal laboratory equipment. The detection limit of the *thrA*-LAMP assay using the LPTV boiling method for DNA extraction was compared to using purified templates obtained from the DNeasy^®^ Blood & Tissue Kit. Performing the LPTV boiling method, *thrA*-LAMP showed a lower detection limit (1.08 CFU/reaction) compared to DNA extraction by the DNeasy^®^ Blood & Tissue Kit (53.8 CFU/reaction). Although the absence of a purification step can negatively affect the detection limit, since possible inhibitors are not eliminated [43,44], the *thrA*-LAMP assay proved to be robust against potentially interfering components. Previous studies have already shown that thermal cell disruption using a boiling method could achieve higher DNA yields than commercial extraction kits based on enzymatic cell treatment [45]. The lower DNA yields of silica membrane-based DNA extraction methods could also be the consequence of insufficient DNA retention [46]. In consideration of these results, the LPTV boiling method was preferred for DNA extraction to gain the highest possible sensitivity from *thrA*-LAMP. In general, comparison of cell-based detection limits between previously published *S. suis*-LAMP assays was limited by the availability of data, differences in DNA extraction methods regarding washing steps and boiling time, as well as primer set design. Using the LPTV boiling method, the *thrA*-LAMP was less sensitive than comparable *S. suis*-LAMP assays (Meng et al.: 18.4 CFU/mL, Arai et al.: 58.3 CFU/mL, Huy et al.: 100 CFU/mL) [22,23,29]. However, because the detection limit for none of these LAMP assays was determined in a spiking experiment, the application-relevant sensitivity in the presence of the sample matrix remains unknown [22,23,29].

Spiking experiments were carried out to evaluate the detection limit of the *thrA*-LAMP in the presence of relevant tissue matrices. The effect of impure and purified templates on the detection rate of LAMP was assessed by comparing the LPTV boiling method and the DNeasy^®^ Blood & Tissue Kit for DNA extraction. Examining unpurified brain templates generated by the LPTV boiling method, *thrA*-LAMP showed a superior detection limit compared to real-time PCR. The missing signal of the internal positive control of the PCR-negative samples indicated the presence of inhibitors as described in the manufacturer’s manual. So far, an interfering effect on DNA amplification has only been described for native cerebrospinal fluid but not for the brain tissue itself [47]. Furthermore, boiling cerebrospinal fluid has been found to be insufficient for removing potential PCR inhibitors, especially when contaminated with blood [47,48]. The inhibitory effect of the brain tissue could have been caused by lipids or small amounts of residual blood [27,43]. The superior resistance of the *thrA*-LAMP assay against containing molecular inhibitors, highlights the robustness of the LAMP method over real-time PCR [26,27,28]. Using the DNeasy^®^ Blood & Tissue Kit for brain swabs, *thrA*-LAMP showed a 10-fold improved detection limit. This is most likely due to the elimination of potentially inhibiting brain tissue components during kit purification [49]. The ability of comparable silica membrane-based DNA extraction kits to generate high-purity eluates from brain tissue was previously demonstrated [50]. In the case of spiked joint swaps, the LPTV boiling method led to a lower detection limit of *thrA*-LAMP compared to brain swabs. Inhibitory matrix effects of sterile synovial fluid components, such as heme or glycoproteins with acidic polysaccharides, have already been described for PCR and real-time PCR [51,52]. Contrary to the spiked brain swabs, the detection limit of *thrA*-LAMP did not improve after the elimination of potential inhibitors using the DNA extraction kit. Surprisingly, *thrA*-LAMP showed a 10-fold higher detection limit compared to real-time PCR when examining impure joint templates, despite PCR being referred to as the more susceptible method [26,27,28]. This result suggests that this assumption does not apply universally and that the performance of both methods depends on the matrix under investigation. In summary, the spiking experiment showed the most promising results for the detection of *S. suis* in brain and joint swabs by *thrA*-LAMP using the LPTV boiling method for DNA extraction. By using the DNeasy^®^ blood and tissue kit, the detection limit of *thrA*-LAMP could be improved for brain swabs but not for joint swabs.

Diagnostic quality criteria of *thrA*-LAMP were determined using cultural investigation and real-time PCR combined with the DNeasy^®^ blood and tissue kits as reference methods. The *thrA*-based LAMP assay achieved high sensitivity (88%) and specificity (95.83%) applying the LPTV boiling method on brain swabs using cultural investigation as reference method. When real-time PCR served as a reference method, the sensitivity of *thrA*-LAMP decreased to 76.67%, but specificity increased to 100%. These differences between reference methods can be explained by the ability of real-time PCR to detect DNA of non-viable bacteria in culture-negative samples, resulting in higher numbers of true-positive samples [53]. Similar observations were made in previous studies in which PCR detected a higher number of *S. suis*-positive cerebrospinal fluid samples compared to culturing [54,55]. Since all clinical specimens were collected from animals with clinical or pathological evidence of *S. suis*-infection, it is possible that animals were treated with antibiotics prior to sampling, resulting in a reduction of vital bacteria and an unsuccessful cultural investigation [54]. Furthermore, it should be noted that about 45% of culturally negative brain and about 88% of culturally negative joint samples originated from organs with inflammatory findings (Appendix A). The formation of biofilms in infected tissues as a strategy to evade host immunity has been demonstrated for *S. suis* [56,57] and was described as a reason for false negative results in cultural testing in both veterinary and human medicine [58,59]. Application of *thrA*-LAMP in combination with the LPTV boiling method on joint swabs showed unexpectedly poor sensitivity, regardless of the viewed reference method. Surprisingly, and contrary to the results of the spiking experiment, LAMP showed a strong improvement in diagnostic sensitivity when clinical joint swabs were processed by kit extraction. Similarly, Kuipers et al. previously showed that silica membrane-based DNA extraction of synovial fluid samples from patients with rheumatoid arthritis obtained higher PCR sensitivity compared to simpler heat-based methods [51]. While only healthy joints were sampled for spiking experiments, field samples were mostly obtained from joints showing evidence of inflammation (85%, Appendix A). Thus, it is possible that the products of inflammation could have an additional inhibitory effect, which could not be accounted for in the spiking experiment. Larger quantities of cell debris resulting from bacterial and host cells in the context of inflammation could have increased the previously described inhibitory effect [43,48]. Furthermore, the resulting higher amounts of released leukocyte or host DNA could be involved in the inhibitory effect proposed for the inflammation [60,61,62]. Both brain and joint field samples were largely obtained from inflamed organs. The difference in sensitivity observed for clinical brain and joint samples extracted by the LPTV boiling method could be due to the higher proportion of inflamed joints (85%, Appendix A) compared to brains (55%, Appendix A) that were sampled. Also, tissue-dependent differences in immune responses, such as the frequency of certain immune cells like plasma cells, could be responsible for the varying sensitivities observed for the targeted organs [63,64]. An inflammation-induced increase in synovial IgG levels by plasma cells could have been responsible for tissue-specific inhibition, which was further enhanced by boiling together with target DNA during the LPTV boiling method [65,66,67,68]. Eliminating these potential inhibitors by applying the DNeasy^®^ Blood and Tissue kit could be the reason for the improvement of *thrA*-LAMP sensitivity observed for clinical joint swabs [51,52]. Many *S. suis*-LAMP assays have not been evaluated against a defined reference method, making it difficult to evaluate the diagnostic quality criteria of *thrA*-LAMP [22,23,29]. Only the *cps2J*-LAMP of Zhang et al. was compared to a real-time PCR targeting the same gene. In that study, examination of 66 clinical samples of pigs and humans, including nasal swabs, blood serum, and cerebrospinal fluid, revealed 96.3% sensitivity and 100% specificity of *cps2J*-LAMP, showing similar results to *thrA*-LAMP using kit extraction and real-time PCR as reference methods [21]. Finally, all isolates recovered from field samples were correctly identified by the *thrA*-LAMP assay in combination with the LPTV boiling method. Thus, this method could be used for rapid confirmation of *S. suis*-suspected colonies as part of a cultural investigation.

## 5. Conclusions

The *thrA*-based LAMP assay established in this study allows specific identification of *S. suis*, including the variety of clinically relevant serotypes. The *thrA*-LAMP was found to be suitable for rapid detection of *S. suis* from brain swabs using the LPTV boiling method for extracting DNA. This easy-to-perform assay requires only a little equipment, making it well suited for field diagnostics. Combining the *thrA*-LAMP with the DNeasy^®^ Blood and Tissue Kit increases sensitivity and allows application to brain and joint swabs. In future field studies, *thrA*-LAMP and LPTV boiling method should be optimized for samples taken from living pigs, such as heparinized blood, synovia, cerebrospinal fluid, and tonsillar swabs, in order to avoid the necessity of a necropsy. Future validation of the *thrA*-LAMP assay regarding human cerebrospinal fluid and synovial punctates would supplement a procedure applicable for intra vitam diagnostics of exposed humans.

## Figures and Tables

**Figure 1 microorganisms-11-02447-f001:**
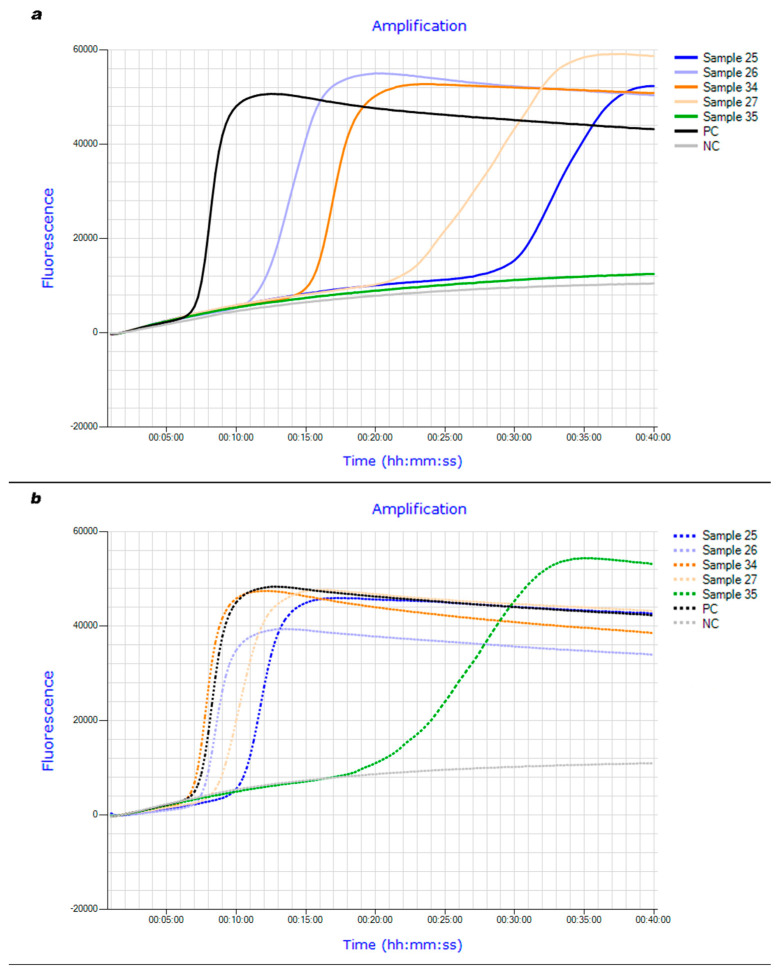
(**a**): Fluorescence curves of *thrA*-LAMP examining brain field samples after DNA extraction using LPTV boiling method. (**b**): Fluorescence curves of *thrA*-LAMP examining brain field samples after DNA extraction using DNeasy^®^ Blood & Tissue Kit.

**Table 1 microorganisms-11-02447-t001:** LAMP Primers used in this study targeting the *thrA* gene (Acc No.: DQ205250.1) of *Streptococcus* (*S.*) *suis*.

Primer	Nucleotide Sequence (5′-3′)	Position
*thrA*-F3	TTCACAGCTAACAGCATGAA	1–20
*thrA*-B3	AATGCGAGCATTTCCTGG	208–225
*thrA*-FIP (F1c + F2)	AGCTGCTAAGAATGCGTCGTAA- ATCGCAGACAGCATCATC	81–102 25–42
*thrA*-BIP (B1c + B2)	TTTTACCCACAGAGGATTGCCT- GGAGCTGACAATAATACCAGC	138–159 184–204
*thrA*-LF	TCATCAATTGGTAGGCTGGC	49–68
*thrA*-LB	GCACGCTATGTCCATCCTAA	160–179

**Table 2 microorganisms-11-02447-t002:** Inclusivity testing of different serotypes and corresponding virulence-associated factor genes of *S. suis* isolates by *thrA*-LAMP.

	Virulence-Associated Factor Genes			No. of Isolates/ No. of Isolates Detected by *thrA*-LAMP
	*mrp*	*sly*	*epf*	
Non- typable	−	−	−	11/11
		+	−	11/11
	+	−	−	4/4
		+	−	10/10
Serotype 1	−	+	−	2/2
	+	+	−	5/5
			+	5/5
Serotype 2	−	+	−	2/2
			+	2/2
	+	−	−	4/4
		+	−	2/2
			+	3/3
Serotype 3 ^a^	+	−	−	1/1
Serotype 4	−	+	−	2/2
	+	−	−	1/1
		+	−	10/10
Serotype 5 ^a^	+	+	−	1/1
Serotype 7	−	−	−	1/0
		+	−	1/1
	+	−	−	10/10
		+	−	2/2
Serotype 8 ^a^	−	−	−	1/1
Serotype 9	−	−	−	1/1
		+	−	4/4
	+	−	−	1/1
		+	−	6/6
Serotype 15 ^a^	−	−	−	1/1

^a^ serotype confirmed by external laboratory, Result of the multiplex PCR described above: non-typable *S. suis.* “+”: present; “−”: not present.

**Table 3 microorganisms-11-02447-t003:** Exclusivity testing of non-target species by *thrA*-LAMP.

Non-Target Strain (Incl. Reference No.)	Detection Time	Annealing Temp.	No. of Isolates
*Actinobacillus equuli* ssp. *equuli*	−	−	1
*Actinobacillus pleuropneumoniae* Serotype 2	−	−	1
*Actinobacillus pleuropneumoniae* Serotype 4 (ATCC 33378)	−	−	1
*Actinobacillus pleuropneumoniae* Serotype 6 (ATCC 33590)	−	−	1
*Actinobacillus pleuropneumoniae* Serotype 9	−	−	1
*Actinobacillus porcinus*	−	−	1
*Actinobacillus suis*	−	−	1
*Actinomyces hyovaginalis*	−	−	1
*Aeromonas hydrophila* (DSM 30187)	−	−	1
*Bordetella bronchiseptica*	−	−	1
*Brachyspira hyodysenteriae*	−	−	1
*Brachyspira pilosicoli*	−	−	1
*Campylobacter coli*	−	−	1
*Campylobacter jejuni*	−	−	3
*Clostridium perfringens* Type B	−	−	1
*Clostridium perfringens* Type C	−	−	1
*Clostridium perfringens* Type D	−	−	1
*Clostridium perfringens* Type E	−	−	1
*Coagulase-negative Staphylococcus* sp.	−	−	7
*Enterococcus durans*	−	−	1
*Enterococcus faecalis* (ATCC 29212, NCTC 8727, DSM 13591, DSM 2570)	−	−	4
*Enterococcus faecium* (DSM 25389, DSM 25390, DSM 2918)	−	−	3
*Enterococcus hirae* (incl. DSM 3320)	−	−	2
*Enterococcus* sp.	−	−	2
*Erysipelothrix rhusiopathiae*	−	−	1
*Escherichia coli* (incl. DSM 1103, DSM 22665, DSM 22311, DSM 22316)	−	−	9
*Escherichia coli* O138:K81	−	−	1
*Glaesserella* (*Haemophilus*) *parasuis*	−	−	1
*Glaesserella* (*Haemophilus*) *parasuis* Serotype 1/2	−	−	1
*Glaesserella* (*Haemophilus*) *parasuis* Serotype 4	−	−	1
*Glaesserella* (*Haemophilus*) *parasuis* Serotype 5/12	−	−	1
*Glaesserella* (*Haemophilus*) *parasuis* Serotype 7	−	−	1
*Glaesserella* (*Haemophilus*) *parasuis* Serotype 8	−	−	1
*Glaesserella* (*Haemophilus*) *parasuis* Serotype 13	−	−	1
*Histophilus somni* (*Haemophilus somnus*)	−	−	1
*Klebsiella pneumoniae* (incl. NCTC 13465)	−	−	2
*Listeria monocytogenes* (incl. DSM 19094)	−	−	2
*Mannheimia haemolytica*	−	−	1
*Micrococcus luteus*	−	−	1
*Pasteurella multocida*	−	−	2
*Pseudomonas aeruginosa* (incl. DSM 939)	−	−	3
*Salmonella* Derby	−	−	1
*Salmonella* Enteritidis	−	−	2
*Salmonella* Infantis	−	−	1
*Salmonella* Newport	−	−	1
*Salmonella* Typhimurium (incl. DSM 19587)	−	−	2
*Serratia marcescens*	−	−	1
*Staphylococcus aureus* (incl. DSM 18597, DSM 799)	−	−	6
*Staphylococcus aureus* (MRSA)	−	−	1
*Staphylococcus aureus* (MSSA)	−	−	1
*Staphylococcus chromogenes* (incl. ATCC 43764)	−	−	2
*Staphylococcus epidermidis* (DSM 1798)	−	−	1
*Staphylococcus hyicus*	−	−	2
*Staphylococcus succinus* ssp. *succinus*	−	−	1
*Staphylococcus xylosus* F	−	−	1
*Streptococcus agalactiae* (incl. ATCC 13813)	−	−	3
*Streptococcus dysgalactiae*	−	−	2
*Streptococcus equi* ssp. *equi*	−	−	1
*Streptococcus equi* ssp. *zooepidemicus*	−	−	1
*Streptococcus* sp. (β-hemolytic)	−	−	10
*Streptococcus thermophilus* (CCUG 21957)	−	−	1
*Streptococcus uberis*	−	−	2
*Trueperella abortisuis*	−	−	1
*Trueperella pyogenes*	−	−	1
*Yersinia enterocolitica* (DSM 11502)	−	−	1
*Yersinia pseudotuberculosis* (DSM 8992)	−	−	1

ATCC: American Type Culture Collection; CCUG: Culture Collection University of Gothenburg; DSM: German Collection of Microorganisms and Cell Cultures; NTCC: National Collection of Type Cultures; “−”: no signal.

**Table 4 microorganisms-11-02447-t004:** Total number of positive signals achieved by *thrA*-LAMP and real-time PCR using LPTV-boiling method and the DNeasy^®^ Blood & Tissue Kit to extract DNA from spiked brain and joint samples.

		CFU/Swab	1–10	10–10^2^	10^2^–10^3^	10^3^–10^4^	10^4^–10^5^	10^5^–10^6^	10^6^–10^7^	10^7^–10^8^
**Tissue**	**Extraction**	**Detection**	**No. of Positive Signals**							
Brain	LPTV boiling method	LAMP	0/3	0/3	0/3	2/3	3/3	3/3	3/3	3/3
		real-time PCR	0/3	0/3	0/3	0/3	0/3	0/3	0/3	1/3
	DNeasy^®^ Blood & Tissue Kit	LAMP	0/3	0/3	0/3	3/3	3/3	3/3	3/3	3/3
		real-time PCR	0/3	3/3	2/3	3/3	3/3	3/3	3/3	3/3
Joint	LPTV boiling method	LAMP	0/3	0/3	1/3	3/3	3/3	3/3	3/3	3/3
		real-time PCR	1/3	2/3	3/3	3/3	3/3	3/3	3/3	3/3
	DNeasy^®^ Blood & Tissue Kit	LAMP	0/3	0/3	0/3	2/3	3/3	3/3	3/3	3/3
		real-time PCR	1/3	1/3	3/3	3/3	3/3	3/3	3/3	3/3

**Table 5 microorganisms-11-02447-t005:** Diagnostic quality criteria of *thrA*-LAMP using LPTV boiling method and DNeasy^®^ Blood & Tissue Kit for DNA extraction of brain swabs with cultural investigation as reference method.

	*thrA*-LAMP	
	**LPTV boiling method**	**DNeasy^®^ Blood & Tissue Kit**
SEN (%)	88.00 (75.26–100)	100 (100–100)
SPE (%)	95.83 (87.84–100)	87.50 (74.27–100)
PPV (%)	95.65 (87.32–100)	89.29 (77.83–100)
NPV (%)	88.46 (76.18–100)	100 (100–100)
Kappa (κ)	0.8369 (0.6843–0.9896)	0.8772 (0.7436–1.0)

SEN: Sensitivity; SPE: Specificity; PPV: positive predictive value; NPV: negative predictive value; 95% confidence interval in parenthesis.

**Table 6 microorganisms-11-02447-t006:** Diagnostic quality criteria of *thrA*-LAMP using LPTV boiling method and DNeasy^®^ Blood & Tissue Kit for DNA extraction of brain swabs with real-time PCR as reference method.

	*thrA*-LAMP	
	**LPTV boiling method**	**DNeasy^®^ Blood & Tissue Kit**
SEN (%)	76.67 (61.53–91.80)	93.33 (84.41–100)
SPE (%)	100 (100–100)	100 (100–100)
PPV (%)	100 (100–100)	100 (100–100)
NPV (%)	73.08 (56.03–90.13)	90.48 (77.92–100)
Kappa (κ)	0.7182 (0.5327–0.9036)	0.9157 (0.8016–1.0)

SEN: Sensitivity; SPE: Specificity; PPV: positive predictive value; NPV: negative predictive value; 95% confidence interval in parenthesis.

## Data Availability

The data presented in this study are available in the presented article or Appendix A.

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
