# Peer review of "Rapid Diagnostic of Streptococcus suis in Necropsy Samples of Pigs by thrA-Based Loop-Mediated Isothermal Amplification Assay"

_microorganisms, 2023, doi:10.3390/microorganisms11102447_

Round 1

Reviewer 1 Report

Streptococcus (S.) suis is a serious threat to the pig industry, the author developed a rapid diagnostic method of S. suis with high specificity and sensitivity. This study is helpful for the onsite diagnosis of S. suis infection. But this method has not been used to test the field samples, if it possible, it is better to verify this method by field samples. Furthermore, the author also need do some minor revision according to the suggestions.

1.Line 20, the author mentioned thattwo different DNA extraction methods were compared, the author should mentioned extract DNA from which material.

2.Line 140, there is mistake in quotation mark.

3.Line 173, the author should illustrate in detail how to select the isolates.

4.Line 91-117, Line 139-163, these paragraph were too long, it is better to separate.

5.Line 200, replace was done by was performed.

6.Line 209, how to prepared the artificially contaminated brain and joint samples, describe it in detail.

7.Line 248-279, this paragraph is too long, it is better to separate. There is no logicality in this paragraph, the author should organize it carefully.

8.Line 259-260, this sentence need to provide reference.

9.It is better to verify this method by field samples.

Quality of English Language is fine. But some paragraph is too long, the

logicality was not clearly.

Reviewer 2 Report

The reviewed manuscript is dedicated to the design and validation of LAMP-based assay detecting Streptococcus suis, a dangerous pathogen causing infections in pigs. The manuscript is detailed and well-written. Presented results are interesting for scientists, specializing on the field of molecular diagnostics. However, a number of issues needs to be addressed before publication.

1.      Where it is possible, authors are encouraged to provide fluorescent curves and/or Tt/Cq values for real-time experiments. It would help to assess time for analysis and technical variability.

2.      3.5. Diagnostic Quality Criteria for Detection of S. suis in Brain and Joint Swabs — authors are encouraged to present results for joints testing in tables as it was done for brain samples.
